# Effect of Physical Activity/Exercise on Oxidative Stress and Inflammation in Muscle and Vascular Aging

**DOI:** 10.3390/ijms23158713

**Published:** 2022-08-05

**Authors:** Mariam El Assar, Alejandro Álvarez-Bustos, Patricia Sosa, Javier Angulo, Leocadio Rodríguez-Mañas

**Affiliations:** 1Fundación para la Investigación Biomédica del Hospital Universitario de Getafe, 28905 Getafe, Spain; 2Centro de Investigación Biomédica en Red de Fragilidad y Envejecimiento Saludable (CIBERFES), Instituto de Salud Carlos III, 28029 Madrid, Spain; 3Servicio de Histología-Investigación, Unidad de Investigación Traslacional en Cardiología (IRYCIS-UFV), Hospital Universitario Ramón y Cajal, 28034 Madrid, Spain; 4Servicio de Geriatría, Hospital Universitario de Getafe, 28905 Getafe, Spain

**Keywords:** exercise, physical activity, oxidative stress, inflammation, aging, muscle, cardiovascular system

## Abstract

Functional status is considered the main determinant of healthy aging. Impairment in skeletal muscle and the cardiovascular system, two interrelated systems, results in compromised functional status in aging. Increased oxidative stress and inflammation in older subjects constitute the background for skeletal muscle and cardiovascular system alterations. Aged skeletal muscle mass and strength impairment is related to anabolic resistance, mitochondrial dysfunction, increased oxidative stress and inflammation as well as a reduced antioxidant response and myokine profile. Arterial stiffness and endothelial function stand out as the main cardiovascular alterations related to aging, where increased systemic and vascular oxidative stress and inflammation play a key role. Physical activity and exercise training arise as modifiable determinants of functional outcomes in older persons. Exercise enhances antioxidant response, decreases age-related oxidative stress and pro-inflammatory signals, and promotes the activation of anabolic and mitochondrial biogenesis pathways in skeletal muscle. Additionally, exercise improves endothelial function and arterial stiffness by reducing inflammatory and oxidative damage signaling in vascular tissue together with an increase in antioxidant enzymes and nitric oxide availability, globally promoting functional performance and healthy aging. This review focuses on the role of oxidative stress and inflammation in aged musculoskeletal and vascular systems and how physical activity/exercise influences functional status in the elderly.

## 1. Introduction

Social and medical advancements achieved during the 20th century led to a great increase in people’s lifespans, doubling the life expectancy worldwide [1]. This trend will be maintained at least during the first half of the present century, which will result in further growth in both the proportion of older people and the proportion of those achieving long longevities. The group of people over the age of 80 is expected to experience a threefold increment [2]. These demographic changes are accompanied by relevant changes in epidemiology that should lead to changes in the clinical field, taking into account the new challenges that this population provides for health and social systems, which are quite different from those of the younger (adult) populations [3].

This challenge for society in general and for health systems in particular means that the aim of these systems is not only improving lifespan but most importantly improving healthspan, thus moving the focus from the quantity of life (life expectancy) to the quality of life, i.e., to function. Aging is characterized by several highly prevalent changes, including an increase in morbidity and a decrease in functional performance which, although linked, are two separate conditions [4,5]. In this sense, functional performance in older people is the most strongly related factor to quality of life and the risk of hospitalization, permanent institutionalization, use of health and social resources, and death [3]. In fact, the World Health Organization (WHO) has recognized the true relevance of function and the components involved in its preservation or deterioration in healthy aging [6]. The WHO defines healthy aging as “the process of developing and maintaining the functional ability that enables wellbeing in older ages”. This means that the elderly’s health status is determined by functional status rather than morbidity, since, according to this definition, older persons with multiple diseases may enjoy a healthy aging process if they maintain functional ability.

In the aging-related trajectory potentially leading to disability, frailty arises as an indicator of functional impairment, with important possibilities for reversion by adequate intervention. Frailty is a geriatric syndrome characterized by a reduced functional reserve, resulting in increased vulnerability to stressors and a limited capacity to maintain homeostasis. Frail elders present poor performance in functional tasks and have an increased risk for negative health outcomes, including falls, institutionalization, mobility impairment and disability, hospitalization and mortality. Frailty is the consequence of the interaction between the aging process and some chronic diseases and conditions that are prevalent in the elderly [7]. This clinical syndrome is characterized by a progressive decline in multiple body systems that are associated with high vulnerability to stressors, which results in a dysregulation of multiple physiological systems [8], including skeletal muscle and the cardiovascular system [9,10]. The musculoskeletal system plays a key role in the decline in muscle strength and functional capacity in the older people. In this sense, sarcopenia, defined as lowered skeletal muscle mass and reduced skeletal muscle strength, is clearly related to the frailty phenotype, although sarcopenia is not always present in frail elders [11], pointing to the existence of several phenotypes of frailty [12]. However, as mentioned, frailty is a multisystem manifestation and, in addition to musculoskeletal system, alterations in other organ systems are also related to functional decline and frailty. In this sense, the aging of the cardiovascular system also affects the functional outcome in the elderly. Cardiovascular health in midlife has been shown to determine the frailty phenotype in later life, favoring a robust functional status [13]. On the other hand, large-population studies revealed an increased risk for myocardial infarction and stroke in frail older people [14]. In fact, the muscular and cardiovascular systems do not stand isolated one from each other, since interactions in the course of aging between both systems seem to occur. Indeed, sarcopenia is more prevalent among cardiovascular disease (CVD) patients and is related to markers of vascular health such as arterial stiffness and coronary artery calcifications [15].

### Oxidative Stress and Inflammation, Constituents of the Background of Aging

Although free radical theory failed to completely explain the aging process, the outstanding role of oxidative damage in the aging-related decline of function is widely accepted [16]. Oxidative damage at the molecular and cellular levels is the result of an imbalance between oxidant and antioxidant processes in favor of the former. Reactive oxygen species (ROS) are key players in the deleterious impact caused by oxidative stress. However, ROS are fundamental for acting as signaling molecules to generate hormetic responses by triggering defense mechanisms, while they cause oxidative damage when generated in excess without modulation. This dual activity could underlie the inconsistent clinical outcomes obtained with antioxidant supplementation [17,18,19]. Analogously, inflammation is a physiological response of the organism to harmful stimuli, but becomes deleterious when the inflammatory process becomes persistent, leading to low-grade chronic inflammation that causes tissue damage and impairs adequate acute inflammatory response. Oxidative stress and inflammation are two closely related processes, and their interdependence is consistently documented [20]. In this sense, ROS/reactive nitrogen species (RNS) can initiate an intracellular proinflammatory cascade [21], while the rise in inflammatory cytokines fuels oxidative stress, creating a vicious cycle [22]. 

Oxidative stress and low-grade chronic inflammation have been demonstrated to play an important role in the hallmarks of aging [23]. However, they are related to an unsuccessful aging outcome rather than to the aging process by itself [24]. In this sense, oxidative stress and inflammation have been shown to be associated with aging-related diseases and frailty [25].

Globally, oxidant damage and inflammation would form a background that promotes functional decline in different tissues and organs. The clinical manifestations of this situation in the form of aging-related diseases and conditions such frailty can be determined by the resilience of the specific systems. In this context, frailty will be manifested when a multisystemic failure occurs that results in a condition prone to disability and mortality [26] (Figure 1).

Aging-associated oxidative damage and chronic inflammation that occur at the cellular/tissue level form a background that promotes the functional decline in cardiovascular system and skeletal muscle [7]. The alterations in cardiovascular territory are characterized by the presence of endothelial dysfunction and increased arterial stiffness [27]. Meanwhile in skeletal muscle a decrease in muscle function and strength is associated with aging. A close interaction between both systems exists in which myokine alteration and reduced blood flow seem to play a role. Frailty arises as the clinical manifestation of this multisystemic failure and results in a condition prone to disability [26]. Physical activity/exercise interventions have been proven to reduce oxidative stress and inflammation at the cellular/tissue level, to limit cardiovascular and skeletal muscle alterations and to reduce the risk of unsuccessful functional outcomes [9]. 

In the pathway leading from robustness to frailty and disability, several approaches have been assessed. Among them, physical exercise is probably the most successful, as has been shown in several randomized controlled trails [28,29,30], usually as part of multimodal combinatory interventions and including the prevention/reversion of frailty [9]. In fact, it has been suggested that avoiding the prescription of exercise programs to older people could be considered unethical [31]. However, the specific mechanisms driving the effects of physical activity/exercise on aged skeletal muscle and cardiovascular systems have not been completely elucidated. This knowledge would help clinicians to adapt the intervention to the specific condition of the older subject. 

This review will focus on the role of oxidative stress and inflammation as drivers of frailty and adverse functional outcomes in aging, with emphasis on the musculoskeletal and vascular systems and on how physical activity and exercise might influence the functional status in the elderly. The analysis of different exercise intervention modalities and some insight into the mechanisms involved in physical activity/exercise impact on muscle and cardiovascular systems will also be provided.

## 2. Aging-Related Changes in Skeletal Muscle and Cardiovascular System

The aging process is associated with different alterations that occur at the muscular and vascular level, leading to an increased risk of morbimortality and functional decline in older people. We will discuss the principal mechanisms implicated in skeletal muscle and vascular age-impairments, with special emphasis on those related to oxidative stress and inflammation. 

### 2.1. Skeletal Muscle 

Skeletal muscle represents the most abundant tissue in animals and humans, comprising up to 50% of their body mass. Skeletal muscle is of critical importance for general health. Indeed, it plays key roles in posture, mobility, thermogenesis, and whole-body glucose homeostasis [32]. 

During aging, there is a reduction in muscle mass and strength that causes a decrease in the ability to carry out activities of daily living, which produces a shift towards a dependent lifestyle for aging people [9,33], characterized by an increase in disability, the number of falls, hospitalization and mortality [34,35]. From the age of 30 years onwards, muscle mass tends to decline at a ratio of 1% per year [36]. This percentage of decline is indeed higher in people older than 60 years [37]. As a consequence, by the age of 70, skeletal muscle strength is 20–40% lower than that of young people [38], leading to a loss of function [34]. This age-related condition which combines a loss of skeletal muscle mass and function is called sarcopenia [37]. Sarcopenia is a multifactorial and complex phenomenon whose underlying mechanisms are not clearly defined. Specific aging-related changes in the muscles involve a decreased cross-sectional area (CSA) of skeletal muscle due to a switch in the fiber type [38], intramyocellular lipid accumulation and, at later stages, fibrotic tissue [36,39], a decrease in the number of satellite cells [40] and neuromuscular degeneration [41]. At the cellular and molecular levels, sarcopenic muscle is characterized by anabolic resistance, mitochondrial dysfunction, chronic inflammation, and increased oxidative stress [40,42,43,44]. 

Due to the importance of skeletal muscle for healthy aging, there is a significant need for an increased understanding of the mechanisms underlying the changes in skeletal muscle structure and function contributing to frailty and sarcopenia. The main factors involved in skeletal muscle age-impairments are anabolic resistance, mitochondrial dysfunction, oxidative stress, and inflammation. 

#### 2.1.1. Anabolic Resistance

The loss of muscle mass during aging has been related to a state of anabolic resistance that generates an imbalance between protein synthesis and degradation in response to a stimulus [37,41]. There is a decrease in protein synthesis pathways and an increase in muscle atrophy-inducing factors [40,44], leading to a decrease in muscle mass that significantly affects mobility [9,45].

The best-defined anabolic pathways leading to protein synthesis in the muscle involve the mammalian target of rapamycin (mTOR), the serine/threonine kinase Akt/protein kinase B (PKB), hormones such as insulin-like growth factor-1 (IGF-1) and insulin [44]. Insulin/IGF-1 signaling is impaired with aging, mainly due to insulin resistance and reduced levels of IGF-1 [9]. In this regard, recent findings have shown a direct, longitudinal relationship between insulin resistance, assessed through HOMA-IR, and the risk of developing frailty in older people without diabetes [46]. IGF-1 triggers the intracellular signaling pathway, leading to the sequential activation of phosphoinositide3-kinase (PI3K) and Akt, which results in the downstream activation of mTOR, which subsequently enhances protein synthesis. However, this mechanism is decreased in aged skeletal muscle [40]. Insulin resistance is part of this anabolic resistance that leads to the loss of skeletal muscle [33,44]. Age-associated insulin resistance in skeletal muscle has been evidenced in both animal and human studies [33]. Increased lactate production in aged individuals could be related to defective pyruvate dehydrogenase phosphorylation associated with insulin resistance [47].

Protein turnover involves the dynamic process of protein synthesis and degradation, and it is a key mechanism for modulating muscle mass and protein quality [48]. Some studies evaluating age-related changes in protein synthesis pathways in humans have shown an aging-related decline in Akt/PKB-mTOR signaling and protein synthesis that contributes to sarcopenia [44]. The breakdown of muscle proteins involves the activation of the ubiquitin proteasome. Although excessive activity of ubiquitin proteasome is associated with reduced myofiber size and age-related muscle atrophy in mice [49], and ubiquitin content has been shown to increase in aged human muscle [50], a decline instead of an increase in ubiquitin proteasome function has been detected in aged muscles [51]. In line with this, it has been suggested that decreased proteasome activity in aging negatively affects protein quality control, causing the accumulation of misfolded and damaged proteins [52], thus compromising muscular function [53]. Other studies have revealed downregulation of the glycolytic metabolism of the fibers in the muscle fibers of elderly individuals compared to young people, especially in type II or glycolytic fibers, as well as a decrease in the GLUT4 transporter, which is responsible for insulin-stimulated glucose uptake. All of these changes may be associated with the loss of CSA and atrophy of type II fibers and loss of muscle mass [33,54]. However, older human skeletal muscle retains the ability to increase GLUT4 with exercise [55]. 

Studies in aged humans have shown an increase in muscle atrophy-related factors. In this sense, increased serum levels of transforming growth factor-ß (TGF-ß) and elevated intramuscular content of myostatin have been detected [44]. The imbalance between protein synthesis and protein degradation, leading to muscle atrophy and fiber loss, is caused by several factors. Among these, the main factors are mitochondrial dysfunction and increased oxidative stress [56,57], which appear as important contributors to the loss of strength and function associated with age [58]. 

#### 2.1.2. Mitochondrial Dysfunction

Multiple studies have linked mitochondrial dysfunction with the development of sarcopenia. Mitochondrial damage accelerates the accumulation of ROS and cellular energy deficiency, particularly in the skeletal muscle, which may contribute to a complex sarcopenic phenotype [34,56]. Mitochondrial quality control (MQC), which includes mitochondrial proteostasis, biogenesis, dynamics and autophagy, is crucial for the maintenance of homeostasis in muscle cells during aging [59,60]

In aging muscle, mitochondrial alterations are associated with downregulation of peroxisome proliferator-activated receptor-gamma coactivator 1-alpha (PGC-1α), as well as with oxidative phosphorylation (OXPHOS) impairment, and mitochondrial morphological changes [56]. These changes have been linked to sarcopenia, poor physical performance, and chronic fatigue [9,61]. 

PGC-1α is considered the main factor in regulating mitochondrial biogenesis, integrity and function, in cooperation with downstream nuclear transcription cofactors, such as nuclear respiratory factor-1 and -2 (NRF-1 and NRF-2) [59,62]. Reduced expression of PGC-1α and mitochondrial transcription factor A (TFAM) and changes in other important factors in the mitochondrial biogenesis have been detected in the skeletal muscle of sarcopenic mice [63]. In addition, the biological activity of AMP-activated protein kinase (AMPK), which plays a key role in transducing metabolic signals to mitochondrial biogenesis, declines with aging in rats [64]. In older humans, muscle atrophy and decreased physical activity, key factors in the development of sarcopenia, have been linked to decreased levels of PGC-1α [61,65]. Furthermore, PGC-1α expression stimulates the expression of antioxidant genes, including heme oxygenase 1 (HO-1) and is increased in tissues and organs with high-energy metabolic loads, e.g., adipose tissue, cardiac, and skeletal muscle [66]. 

Mitochondrial proteostasis plays an essential role in maintaining the balance between synthesis and protein degradation. Alterations in mitochondrial proteostasis during aging may result from the activation of muscle atrophy regulators and/or the inactivation of genes regulating protein production such as mTOR complex 1 (mTORC1), contributing to muscle atrophy [58].

On the other hand, the alteration of autophagy-mediated by mitochondria (mitophagy) in aged skeletal muscle has been associated with muscle wasting and muscle strength reduction [61,67,68]. Mitophagy dysregulation has been denoted as a pathogenic mechanism in muscle atrophy by an increase in LC3-II, p62, and lysosome-associated membrane protein 1 (LAMP1) expression in sarcopenic mice [63]. The downregulated expression of the autophagy mediator, LC3B, has also been detected in muscle from hip-fractured sarcopenic elderly patients [61].

Therefore, many observations in aged animal models and in humans supported the assertion that the dysregulation of mitochondrial dynamics and function was associated with aging-induced skeletal muscle atrophy [34,69]. One of the consequences of this mitochondrial dysfunction may be related to the presence of oxidative stress in aged muscle. It is widely accepted that mitochondria serve as a significant source of oxidants as well as a primary target of oxidative stress. Moreover, evidence shows that both mitochondrial dysfunction and the increase in oxidative stress associated with age are dependent on physical activity [59].

#### 2.1.3. Oxidative Stress

A decrease in the redox balance with aging has been widely shown [34,70]. Oxidative stress has been postulated to play an important role in the outcome of aging at the functional level [7,26]. The imbalance between the production of ROS, RNS and antioxidant defenses in the body has been suggested to be an early biomarker of sarcopenia [37,71]. It is widely accepted that the bulk of ROS produced by muscle contraction are generated by the mitochondrial electron transport chain during normal oxidative respiration [72], although other non-mitochondrial sources of ROS such as NADPH oxidase (NOX) and cyclooxygenase-2 (COX2) seem to significantly contribute to ROS generation in skeletal muscle [7,73]. 

Within physiological levels, ROS can promote host defense mechanisms such as the activation of signaling pathways, including the mitogen-activated protein kinase (MAPK) signaling pathways, NF-kB signaling pathway and Keap1-Nrf2-antioxidant response element (ARE) signaling pathway. These pathways play an important role in various cellular processes such as cell growth, inflammatory response, autophagy or adaptive response for oxidative stress [74]. ROS contribute to age-related deficits in the muscle through increasing damage to cell constituents and through the induction of defective redox signaling [75].

There is an increase in ROS production in skeletal muscle with aging in mice that may contribute to aging-related muscle atrophy [57]. Excessive ROS induces proteolysis by enhancing the ubiquitin-proteasome system, resulting in skeletal muscle atrophy [76] and motoneuron degeneration in aged skeletal muscle [44]. In fact, Cu/Zn superoxide dismutase (SOD1)-knockout mice show accelerated loss of muscle mass and function due to increased ROS generation, resulting in the loss of muscle fibers and sarcopenia [77].

Moreover, the accumulation of ROS/RNS is thought to be a determinant not only of muscle quantity reduction but also of the loss of muscle quality with aging [78]. Aging-induced accumulation of ROS/RNS has been proposed to decrease muscle quality by impairing muscle fiber activation, excitation/contraction coupling and cross-bridge cycling within the myofibrillar apparatus [79]. 

Antioxidant response represents a main feature of the ROS-induced signaling, which, indeed, prevents the persistence of oxidative stress. In this sense, in response to an increase in oxidative stress, the nuclear factor Nrf2 is activated, inducing antioxidant genes transcription [80]. Nrf2 has a critical role in ROS-induced antioxidant response in skeletal muscle and the cardiovascular system, and its downregulation has been implicated in both sarcopenia and CVD [81]. In line with this, Bose and colleagues demonstrated that increased oxidative stress was negatively correlated with a dysfunction in Nfr2 response in aged skeletal muscle, while nutritional supplementation with the Nrf2 activator sulforaphane resulted in improved strength and fatigue resistance in old mice [81]. This concept was supported by other studies with Nrf2 knockout mice, where the absence of Nrf2 was related to an increase in ROS levels and an altered contractile capacity in skeletal muscle [80].

Human studies showed the association of reduced systemic mRNA expression of Nrf2 and three of its target genes (heme oxigenase-2, thioredoxin reductase-1 and superoxide dismutase-2) with the presence of frailty in community-dwelling older adults, supporting animal data and highlighting the role of an adequate antioxidant response in the functional outcome in the elderly. Furthermore, a positive association between Nrf2 gene expression and gait speed has been found [82]. 

The imbalance in ROS production correlates with an increase in inflammatory mediators, such as tumor necrosis factor- α (TNF-α), interleukin-6 (IL-6), NF-kB and C-reactive-protein (CRP) [37], and leads to a chronic inflammatory state that creates a vicious cycle wherein chronic oxidative stress and inflammation feed into each other [83]. Thus, together with oxidative stress, inflammation seems to play a potential role in muscular alterations related to aging. 

#### 2.1.4. Inflammation

Inflammaging is the chronic low-grade inflammatory state present in older people, characterized by increased systemic concentrations of proinflammatory cytokines such as TNF-α, IL-6, and CRP, among others [40]. It has been shown that inflammaging increases the risk of pathologic conditions and age-related diseases, and it is associated with increased skeletal muscle wasting, strength loss, and functional impairments [40,84].

Recent evidence suggests that the persistent elevation of inflammatory cytokines in sarcopenic patients was associated with impaired satellite cell regeneration [85], as well as with reductions in endothelial reactivity and muscle perfusion, leading to anabolic deficiencies and/or excessive muscle proteolysis [84]. Inflammaging also affects the anabolic–catabolic balance in skeletal muscle cells, causing a shift towards catabolism, atrophy and the progression of sarcopenia, which is a major contributor to functional decline and frailty [86].

TNF-α is a major activator of the apoptotic signaling pathway that leads to increased degradation of muscle proteins and results in muscle atrophy [40]. Its impact on muscle may also be related to the inhibition of muscle regeneration by blocking the proliferation and differentiation of muscle stem cells [87]. IL-6 acts as a multifactorial cytokine (pro-inflammatory and anti-inflammatory) depending on the condition [88], but chronic systemic elevation of IL-6 leads to muscle atrophy via blunting muscle anabolism [89]. Both TNF-α and IL-6 induce the activation of NF-kB, which in turn activates multiple genes implicated in inflammation and proteolysis, leading to skeletal muscle loss [7].

Blood levels of TNF-α and IL-6 have been reported to increase 2- to 4-fold in old persons compared with healthy young adults, promoting sarcopenia [90]. For example, cross-sectional analysis from the InCHIANTI study (1020 men and women over 65 years) demonstrated a significant association of IL-6, interleukin-1 receptor (IL-1R), and CRP serum levels with both poor physical performance and reduced muscle strength [91].

Chronic inflammatory state is not only dependent on the increased expression of pro-inflammatory mediators, but also on reduced levels of anti-inflammatory cytokines such as interleukin-10 (IL-10) [92]. This cytokine has been shown to play an important anti-inflammatory role by inhibiting the production of pro-inflammatory cytokines by monocytes [93]. Importantly, mice models with muscle-specific overexpression of IL-10 have been associated with a low level of age-related muscle inflammation and insulin resistance [94]. Furthermore, IL-10 homozygous knockout mice (IL-10^tm/tm^) were identified as a transgenic model of frailty, since they showed a frail phenotype including inflammation and reduced muscle strength [95].

#### 2.1.5. Myokines 

Myokines are defined as cytokines, hormones and other peptides that are produced, expressed and released by muscle fibers in response to muscle contraction [96]. Myokines exert an autocrine function in regulating muscle metabolism in addition to its paracrine/endocrine regulatory function on other organs or tissues including adipose tissue and the heart, among others, and providing a molecular interaction between muscle function and body physiology [88,97]. Myokines regulate several processes associated with physical frailty, including muscle wasting, dynapenia (age-related reduction in muscle strength), and slowness [9]. Multiple studies have reported that myokines mediate exercise-associated anti-inflammatory effects and the reduction in age-related loss of muscle mass and function [96].

During the aging process, the synthesis of the myokine apelin by skeletal muscle is decreased, and its plasma levels also decreases. Apelin is related to the induction of mitochondriogenesis and can reduce inflammation, stimulate regenerative properties, and avoid age-associated muscle wasting [98]. Another myokine, myostatin, is considered a negative regulator of muscle mass, impairing muscle synthesis and augmenting muscle catabolism [88]. Massive muscle hypertrophy is observed in myostatin knockout mice, which show an increase in fiber CSA and fiber number [96]. Follistatin is a myostatin-binding protein that is capable of inhibiting myostatin activity. Although it has been suggested that follistatin could lead to muscle growth [99], contradictory results have been obtained, since an increased serum follistatin level was independently associated with decreased gait speed among community-dwelling older individuals [100]. 

It is important to note the close network between myokines and other molecules such as adipokines (secreted by adipose tissue) or cardiokines (secreted by the heart) [101]. These mediators play a crucial role in homeostatic adaptation and in counterbalancing the systemic chronic low-grade metabolic inflammation in aging and diseases [88]. In this sense, adiponectin (ApN) is a hormone with insulin-sensitizing and anti-inflammatory properties in several tissues, including the skeletal muscle [102]. Its major effector protein in skeletal muscle is the AMPK, a critical cellular energetic sensor [103]. A study in aged mice reported that treatment with an adiponectin receptor agonist (AdipoRon) improved muscle regeneration, muscle function, and physical performance by producing changes in fiber type and by increasing mitochondrial activity [103].

### 2.2. Cardiovascular System

Aging is the main risk factor for CVD, even in the absence of traditional risk factors, while CVD is considered the principal contributor to morbidity and mortality in older populations [104,105]. The aging process is associated with both structural and functional alterations at the vascular level, leading not only to an increase in cardiovascular events in older subjects but also to functional decline, cognitive deterioration, and frailty [9,106]. 

Evidence derived from longitudinal studies has demonstrated that vascular aging is associated with two specific arterial phenotypes: endothelial dysfunction and an increased stiffness of the large elastic arteries [27]. In fact, different researchers have clearly demonstrated the presence of endothelial dysfunction, manifested by impaired endothelium-dependent vasodilation, associated with the aging process in the micro- and macrovasculature of animal models and humans [106,107].

Increased arterial stiffness, another hallmark of vascular aging, is characterized by a decrease in arteries’ elasticity and is manifested by an increase in pulse wave velocity (PWV) [108]. Age-related stiffening of large elastic arteries is primarily attributed to increased levels of matrix metalloproteinase-2 (MMP-2) in animals and humans [109]. Klotho-deficient mice, an animal model of unsuccessful aging, present greater aortic stiffness and blood pressure that are accompanied by reduced elastin levels, as well as elevated MMP-2 and MMP-9 expression when compared to control mice [110,111]. In line with this, MMP-2 knockdown attenuates age-dependent carotid stiffness by blunting elastin degradation and augmenting endothelial nitric oxide synthase (eNOS) bioavailability in mice [112]. 

Importantly, arterial stiffness is linked to endothelial dysfunction [9]. At the same time, endothelial dysfunction plays an important role in the development of atherosclerosis [113]. In fact, increased levels of asymmetric dimethylarginine (ADMA), a marker of endothelial dysfunction, were related to an increase in the risk of frailty in older adults free of arteriosclerosis [114]. Furthermore, previous studies have reported an association between arterial stiffness and skeletal muscle. In this sense, increased arterial stiffness was associated with limited blood flow volume in the lower and upper extremities, lower muscle mass, and diminished physical function, leading to the onset of sarcopenia [10,115].

Several studies have shown a relationship between CVD and reduced muscle mass in young and old people [116]. Some studies linked muscle mass loss with vascular calcifications [117]. In the Melbourne Collaborative Cohort study, older women with severe abdominal aortic calcification showed a decline in handgrip strength [10]. Furthermore, Rodríguez and colleagues confirmed the association between vascular calcification and decreased muscle strength [118]. 

Poor cardiovascular function in the elderly is related to the onset of frailty, and frailty is an adverse prognostic factor in cardiac patients [116]. In line with this, Minn and colleagues reported that high muscle mass might protect against ischemic stroke in community-dwelling adults [119]. Finally, other authors have reported a link between low muscle mass and an increase in CVD mortality in individuals older than 65 years with CVD risk factors [113].

Although the mechanisms responsible for aging-related vascular dysfunction have not been completely elucidated, oxidative stress and chronic low-grade inflammation are considered the major contributors to age-related vascular dysfunction [106].

#### 2.2.1. Oxidative Stress

Vascular oxidative stress is considered a primary mechanism underlying impaired endothelium-dependent vasodilatation and increased arterial stiffness related to aging. An elevated content of several markers of oxidative stress, such as 4-hydroxynonenal (4-HNE) or malondialdehyde (MDA), has been detected in aged arteries [27]. 

At the functional level, ROS and, in particular, superoxide anions react with NO, leading to the formation of peroxynitrite [83] and the subsequent reduction in NO bioavailability. This latter is associated with increased platelet activity, endothelial dysfunction, inflammation, and the initiation, progression, and complications of atherosclerosis [120]. In line with this, increased superoxide anion and peroxynitrite (ONOO^−^) formation was detected in aged human vessels that presented a defect in endothelium-dependent vasodilation [121]. Furthermore, the inhibition of ONOO^−^ with its scavenger, FeTMPyP, normalized vasorelaxation and suppressed nitrotyrosine expression, the footprint of peroxynitrite formation in resistance arteries of aged rats [122]. Moreover, reduced NO bioavailability was associated with arterial stiffness in women older than 65 [27]. 

Uncoupled endothelial nitric oxide synthase (eNOS) is considered an important source of superoxide anion. eNOS uncoupling occurs when the availability of the critical cofactor tetrahydrobiopterin (BH4) is inadequate, leading eNOS to produce superoxide anion instead of NO [109]. In aged rats’ arterioles, reduced levels of BH4 were accompanied by impaired endothelium-dependent vasodilation [123]. Furthermore, in estrogen-deficient postmenopausal women, reduced vascular BH4 seems to be an important contributor to arterial stiffening, related in part to reduced endothelial-dependent vasodilatory tone [124].

Although NOS uncoupling accounts for ROS generation, mitochondrial ROS production has an important role in age-related vascular dysfunction. In the aged vasculature, there is an increase in ROS due to dysfunctional electron transport chain, inhibition of mitochondrial antioxidant enzyme manganese-SOD (SOD2), down-regulation of p66Shc55, and/or impaired Nrf2-mediated antioxidant defense responses [125]. Treatment with mitochondrial-targeted antioxidants improved endothelial function in aged mice [126]. Furthermore, Park and colleagues demonstrated that treatment with mitochondria-targeted antioxidant MitoQ reverses age-related vascular dysfunction in human skeletal muscle feed arteries [127].

NOX enzymes are another major source of oxidative stress in the cardiovascular system [128]. NOX are expressed throughout the vessel wall, including endothelial cells and vascular smooth muscle cells [129]. A large volume of evidence of age-related upregulation of NOX expression or activity has been obtained in aged rats [109]. In fact, in a rodent model of aging, it was observed that age-dependent increases in blood pressure, cardiomyocyte hypertrophy, coronary artery remodeling, and cardiac fibrosis were associated with increased myocardial NOX2 activity [130]. In line with this, the attenuation of NOX activity improved endothelial function in aged coronary arteries [126]. Furthermore, increased mitochondrial NOX4 expression seems to play a causative role in age-related aortic stiffness in hypercholesterolemic mice [128]. 

Vascular aging is not only characterized by increased ROS generation, but also by a dysregulated antioxidant defense. In this sense, the Nrf2 antioxidant defense pathway plays a central role in vascular stress by regulating both cellular DNA repair and the elimination of ROS. Importantly, genetic depletion of Nrf2 exacerbates age-related vascular senescence [131]. A study showed that the increased oxidative stress in the aging heart correlates with Nrf2 dysregulation, and this drop in the protective response was attenuated by the administration of sulforaphane to old mice [81]. Furthermore, short-term pharmacological activation of Nrf2 decreased the age-related impairment of endothelium-dependent and ROS-induced vasodilatation in different vascular territories in rats and humans [132].

#### 2.2.2. Inflammation

In addition to oxidative stress, inflammation stands out as a determinant underlying the mechanisms implicated in vascular aging. In fact, the concept of inflammaging which was adopted by Ferrucci recognizes that the chronic low-grade inflammation milieu observed in older adults contributes to cardiovascular risk [133], and seems to play a role in the development of multiple age-related diseases and conditions, including frailty [134]. However, the pathways linking inflammaging to physiological function and healthspan in humans are not well-known [135]. 

A large volume of evidence has observed an increase in the different systemic markers of inflammation, such as CRP and IL-6, being positively related to aortic stiffness and inversely correlated to endothelial dysfunction in older adults [125]. Additionally, the TNF-α, IL-1β, and NOD-like receptor protein 3 (NLRP3) inflammasome can contribute to age-related progression of hypertension [136]. According to this, a study in NLRP3 KO old mice revealed that the ablation of NLRP3-inflammasome prevented many age-associated changes in the heart, preserved the cardiac function of aged mice and increased lifespan [137]. 

Inflammation involves the activation of the ROS-sensitive, pro-inflammatory transcription factor, NF-κB. This factor is the master regulator of inflammatory molecules including TNF-α, interleukins (IL-1β, IL-2, and IL-6), chemokines (IL-8 and RANTES), adhesion molecules (ICAM, and VCAM), and enzymes (iNOS and COX-2) [109], and is believed to play a critical role in age-related vascular inflammation. In fact, aged endothelial and smooth muscle cells exhibit significant NF-κB activation [121]. Selective inhibition of NF-κB in the vasculature was shown to improve blood flow regulation and decrease systemic inflammation [126]. Moreover, an age-associated increase in NFκB activity has been directly implicated in arterial dysfunction in older rodents and humans [27]. 

The activation of NF-κB may be evoked by dysfunctional mitochondria [133], indicating the presence of a positive vicious cycle between inflammation and oxidative stress. ROS generation seem to play a key role in the progression of inflammation in blood vessels [10]. Therefore, therapeutic interventions which combine antioxidants and anti-inflammatory activities, such as resveratrol, may have a role in preventing vascular dysfunction in the elderly [109]. In aged mice treated with resveratrol, lower aorta media thickness, lower inflammation, and lower fibrosis and oxidative stress were observed when compared to the control group [138].

Taking into account the involvement of inflammation in the pathogenesis and progression of CVDs, the search for potential biomarkers linked to microvascular dysfunction might help to improve diagnoses, disease progression and therapy response [139]. In this sense, non-coding RNAs have emerged as attractive biomarkers in heart failure, and their potential clinical applications may be feasible in the era of personalized medicine [140].

The age-related chronic low-inflammation state is not only sustained by an increase in inflammatory mediators but also by a reduction in circulating anti-inflammatory cytokines such as IL-10 and adiponectin. This imbalance might exacerbate vascular extracellular matrix remodeling and arterial stiffening [136]. For example, treatment with IL-37, a critical anti-inflammatory interleukin in humans [141], improves vascular endothelial function, endurance exercise capacity, and whole-body glucose metabolism in old mice [142].

It is important to note that there is a crosstalk between some myokines and vascular function. In this sense, higher concentrations of irisin were detected in centenarian people without CVD when compared to young individuals, which could be indirect evidence of the protective role of irisin against the development of CVD. Furthermore, irisin and follistatin-related protein 1 (FSTL1) improve endothelial cell function and arterial relaxation, and protect against endothelial injury and atherosclerosis through the activation of PI3K–Akt and eNOS signaling [113]. 

Improvements in our knowledge about the molecular and cellular mechanisms involved in vascular aging and functional decline as well as their potential interactions provide a growing list of factors that can be targets for specific intervention to prevent or delay the onset of frailty.

## 3. Impact of Physical Activity/Exercise on Aged Skeletal Muscle and Cardiovascular System

Physical activity and exercise training appear as the clearest modifiable determinants of functional outcome in the elderly. This is due in part to their impact on multiple key systems affected by the aging process, including the skeletal muscle as well as the cardiovascular system [9,143]. 

### 3.1. Skeletal Muscle 

Physical inactivity has emerged as one of the greatest threats to the health of the global population [144] and their health systems [145]. Physical inactivity per se has been linked to an increased risk of the incidence of several diseases [146], and is nowadays considered one of the leading causes of preventable death [146,147,148]. 

Although several definitions have been proposed to define physical activity [149], the most used one was published by Caspersen and colleagues [150] as “any bodily movement produced by skeletal muscles that results in energy expenditure”. 

Low physical activity is a domain of frailty [151] and a predictor of mortality by itself in community-dwelling older adults [12]. In this sense, isotemporal substitution analysis showed that 1 h of moderate- to vigorous-intensity physical activity (MVPA) instead of sedentary or light physical activity was associated with higher values of muscle mass, gait speed and grip strength, as well as with a reduction in the risk of being sarcopenic by almost half [152]. This latter indicates that a minimum level of intensity is needed for muscle mass improvement or maintenance. 

Exercise is a subset of physical activity, being planned, structured and repetitive [150], and its benefits have been extensively studied. Exercise improves different parameters in older adults, such as strength [153,154] and physical function [154,155], and reduces disability [28,30], as well as preventing falls and fall-associated injuries [156], demonstrating that lengthy interventions are safe [157]. Even in older adults who meet the sarcopenia criteria, physical exercise significantly improves strength (assessed by grip strength, knee extension and chair test), physical functioning (timed up and go test and gait speed) and muscle mass [158]. 

Nevertheless, it is important to note that not everything is valid when it comes to specifically improving certain parameters, and so defining certain aspects such as frequency, intensity, time, or type of exercise could be key to obtaining the final result. This aspect is particularly relevant in the presence of certain conditions, which can substantially modify the response to exercise [159]. In line with this, the exercise recommendations of WHO for older adults, even those with chronic conditions or disability, include a multicomponent exercise program of at least moderate intensity on 3 or more days per week to improve functional capacity and prevent falls [160]. These recommendations include performing at least 150 min of moderate-intensity or 75 min of vigorous-intensity aerobic exercise spread over 3 or more days per week and performing a balance and strength training program of moderate or higher intensity involving all muscle groups 3 or more days per week [160]. In fact, older adults who meet the guidelines for aerobic exercise, strength exercise, or both, are associated with greatly reduced risk of all-cause mortality, as well as disease-specific mortality, such as hypertension, heart disease, stroke, cancer, or Alzheimer’s disease [161]. 

On the other hand, resistance training has been proposed as the gold-standard treatment to counteract the age-associated wasting of muscle mass, neuromuscular performance and cellular adaptations [162,163,164,165]. In a recent study comparing different strength training frequencies (3 days a week vs. 2) and different intensities (low-load vs. high-load) in adults over 65, it was shown that high-load exercise 3 days per week over a 2-year period of supervised training significantly increased appendicular lean mass when compared to those subjects who performed only 2 days of exercise at light loads. Furthermore, all trained groups showed a similar improvement in their strength [166]. In fact, the combination of aerobic and resistance exercise may be the most effective combination for improving muscle mass and one of the best for improving strength and functional capacity in older adults with sarcopenia [167]. 

One entity that can co-exist alongside sarcopenia is frailty [11]. A recent systematic review examined the evidence on the effect of strength training on muscle strength, physical function, body composition, pre-sarcopenia, sarcopenia, pre-frailty and frailty. Strength training in early and advanced stages of sarcopenia and frailty was highly effective in improving these parameters [168]. 

Alternatively, due to the multiorgan and biological systems that are involved in the frailty syndrome, the inclusion of different types of exercise, such as strength, aerobic, balance or flexibility, could be the best intervention for this type of patient [30,154,169,170,171]. In line with this, recent research has been published with data from the SPRINTT study (Sarcopenia and Physical Frailty in Older People: Multi-component Treatment Strategies) conducted in community-dwelling older adults with a mean age of 78.9 years with physical frailty who also met the sarcopenia criteria. In this study, subjects with moderate physical function, assessed with the Short Physical Performance Battery (SPPB, between 3 and 7 points) who undertook a multi-component exercise program (moderate-intensity physical activity supervised twice a week, and up to 4 times a week at home), together with nutritional counseling, displayed reduced number of disability events. In addition, the difference between the groups with respect to their SPPB score was 0.8 points and 1 point at 24 and 36 months in the intervention group, respectively. Additionally, the loss of grip strength and muscle mass was lower in women in the intervention group [30], demonstrating that exercise can relieve the decline in functional performance, strength and muscle mass associated with aging. 

Muscle power has been linked to physical function in older adults [172,173]. Power exercise has been proposed as an effective intervention to improve cognitive, neuromuscular and physical function in older adults [174,175], and has been suggested for inclusion in the physical exercise recommendations for this population [176,177]. However, while this type of intervention may improve muscle mass in older adults compared to the control, it does not offer further improvements in hypertrophy over conventional training [178]. Furthermore, in a meta-analysis evaluating the effect of training at high versus moderate intentional speeds, no significant improvements were found for either exercise modality [179]. In a recent study by Coelho-Júnior and Uchida [180], the effect of a 16-week training program on functional parameters in prefrail and frail individuals was evaluated. Participants were randomized into a control group, a low-speed strength training group and a high-speed strength training group. Although in both training groups frailty status was reversed and their physical performance was improved notably, different patterns of improvement were observed among resistance training protocols. In fact, high-speed training was the most effective in reversing frailty status, as well as mobility and dual tasks, whereas low-speed training was the most effective in improving strength, power and balance. Additionally, this study reported no differences in blood pressure or heart rate.

### 3.2. Cardiovascular System

Cardiorespiratory fitness (CRF) is the ability of the cardiovascular (heart and blood vessels) and respiratory (lungs) systems to supply oxygen to the musculoskeletal system during sustained physical activity [150]. One of the best studied markers of physical fitness is maximal oxygen uptake (VO_2_ max), increased levels of which decreased the overall mortality rate [181]. In fact, lower VO_2_ max was shown to be reduced in older subjects [55]. Furthermore, it has been hypothesized that elevated levels of VO_2_ max is associated with the improvement or maintenance of different systems or organ functions (including heart and muscle) in middle aged and older subjects [143]. Thus, to evaluate CRF means to assess the proper functioning of heart, blood vessels and lungs, among others. CRF should be assessed in clinical practice due to its association with cardiovascular disease, all-cause mortality, and mortality rates attributable to several clinical conditions [182,183,184,185,186]. During aging, there is a decrease in CRF, but this decline is nonlinear and could be modulated by a physically active lifestyle [187]. 

Physical activity has positive effects on cardiovascular risk factors, such as type 2 diabetes, arterial function, myocardial infarction and heart failure [188]. Recently, data from a large prospective study were published, suggesting that total physical activity reduces the risk of myocardial infarction in women, whereas participation in leisure-time physical activity reduces the risk of myocardial infarction and stroke in men [189]. In addition, Gonzalez-Jaramillo and colleagues [190] looked at physical activity patterns in patients with coronary heart disease. They observed that those who remained active, became active, and those previously active who became inactive had mortality reductions of 50%, 45%, and 20%, respectively, when compared with subjects that remained inactive. For cardiovascular death, those who remained active and increased their activity had a significantly lower risk than those who remained inactive [190]. 

Arterial stiffness is another early marker of cardiovascular disease, which has been associated with muscle mass [191] or frailty [192]. Physical activity, objectively monitored by steps per day, has been shown to be significantly associated with arterial stiffness in adults and older adults [193].

Furthermore, exercise has emerged as a cornerstone in the non-pharmacological treatment of patients with incipient or established hypertension [194,195]. There is moderate evidence that physical exercise improves physical function and quality of life [196], decreases mortality associated with cardiovascular disease [161,196], and decreases systolic and diastolic hypertension, especially aerobic exercise [197,198]. Endurance training attenuates age-related endothelial function in older men [199,200] but not in older women [199]. Nevertheless, after 19 exercise sessions over 10 weeks, an interval aerobic training program with recovery bouts significantly reduced blood pressure values in sedentary older adults, of whom 70% were women [201]. Eight weeks of moderate-intensity exercise significantly improve endothelial function in middle-aged and older individuals (including men and women) with prehypertension or hypertension, irrespective of the exercise-type performed [202]. Furthermore, in adults over 60 years of age with or without hypertension, there are no significant differences between performing exercise in the format of high-intensity interval training (HIIT), intermittent bursts of vigorous-intensity exercise interspersed with periods of low-intensity exercise or rest, and moderate-intensity continuous training [203]. On the other hand, an aerobic exercise program also improved cerebral blood flow in sedentary men aged 60–70, which may explain the beneficial effects of aerobic exercise on executive function and improvements in glucose metabolism [204]. 

Alternatively, strength exercise has been proposed as an effective tool to reduce both systolic and diastolic blood pressure in prehypertensive and hypertensive subjects, being particularly effective in older adults [205]. Strength exercise impacts on functional capacity, muscle strength and mobility in older adults with coronary artery disease [206]. Moreover, a recent network meta-analysis studied the effects of different types of exercise on arterial stiffness, finding that either combined exercise, aerobic exercise, or intervallic training showed significant improvements, with the most effective being combined mind–body interventions (such as Pilates, Tai Chi or Yoga) [207]. However, this study mostly included young subjects and did not consider other types of exercise such as stretching. Stretching was shown to be an effective strategy in a meta-analysis to reduce arterial stiffness, improve endothelial function, and reduce blood pressure, especially diastolic blood pressure [208].

It is important to note that each subject does not respond equally to exercise and may not respond to a particular intervention or in a given way, and that there is a wide inter-individual variability, especially in certain clinical settings or conditions [209,210]. In fact, one study reported alterations in systolic blood pressure, HDL, triglycerides, and insulin levels [211]. However, given the solid evidence in favor of the potential benefits of exercise, not providing it to the aged population may be unethical [31].

## 4. Impact of Physical Activity/Exercise on Oxidative Stress and Inflammation in Muscle and Vascular Aging 

The effects exerted by physical activity/exercise seem to be systemic and there are multiple signaling pathways that are beneficially modulated by exercise [143], including those related to oxidative stress and inflammation. Understanding the different molecular mechanisms through which physical activity and exercise exert their effects on skeletal muscle and vascular functions may help to develop adequate strategies aimed at improving physical performance in elderly subjects and to prevent or even reverse/reduce frailty. Therefore, we will discuss the underlying signaling pathways related to oxidative stress and inflammation which are modulated by physical activity/exercise in the aged skeletal muscle and cardiovascular system. 

### 4.1. Skeletal Muscle 

Skeletal muscle plays a central role in the response to physical activity/exercise. As mentioned above, there is increasing evidence supporting the benefits yielded by exercise and multimodal interventions on the functional status of older people, including the prevention/reversion of frailty [9] and in managing sarcopenia, which is considered one of the substrates of frailty [212]. 

The impact of exercise on skeletal muscle is rather vast and complex. In this sense, the anti-oxidative effects of exercise training are widely accepted (Figure 2). In fact, exercise training was postulated by some authors as an antioxidant [70]. It increases oxidant production by cells which is not only limited to muscle cells but also encompasses endothelial cells associated with contracting muscles. This seems to be a key element of hormesis activation of antioxidant signaling pathways, resulting in an increase in antioxidant capacity [70]. In line with this, a very recent clinical trial has shown that a greater volume of resistance training can promote superior improvement on different oxidative stress biomarkers in older women [213].

It is important to note that the response to exercise is not homogeneous and it depends on its type, duration, and intensity. For instance, single bouts of exercise when exceeding a certain intensity and duration have been shown to increase ROS production by mitochondria and cellular oxidases, leading to cell damage [214]. In contrast, chronic exercise alleviated oxidative stress in aged skeletal muscle [215]. Furthermore, less systemic concentrations of oxidized LDL, considered a marker of oxidative stress, have been detected in peripheral mononuclear cells derived from trained older subjects when compared to sedentary individuals, concomitantly with a lower expression of genes involved in oxidant production [216]. A recent study found that decreased systemic oxidative stress was associated with exercise-induced limb muscle structural and functional adaptations (increased muscle size, pennation angle, muscle strength and exercise capacity) in older individuals with chronic obstructive pulmonary disease [217].

Another target of exercise against oxidative stress is the increased activity of antioxidant response. In this sense, lifelong-trained older subjects showed increased catalase expression in muscle biopsies when compared to the untrained counterparts [218]. Furthermore, Bouzid and colleagues observed a significant increase in blood SOD and glutathione peroxidase (GPX) activities in old subjects after exercise when compared to old sedentary subjects. By contrast, similar antioxidant activities and lipid peroxidation were detected between old active and sedentary young subjects, suggesting that beneficial effects of regular physical activity in antioxidant defense and lipid peroxidation damage could be impaired by the aging process [219].

Despite their well-known deleterious role, ROS have been recognized as key players in initiating an adaptive response in exercising muscles, with H_2_O_2_ playing a central role. In fact, a chronic increase in mitochondrial H_2_O_2_ has been pointed to as being responsible for redox-attenuated adaptation to contractile activity associated with aging [220]. The adaptive response promoted by exercise involves the transcription of different redox sensitive factors including NF-κB, MAPK, and PGC-1α, among others, resulting in enhanced cytoprotective proteins such as SODs, catalase, and heat shock proteins that prevent oxidative damage. This adaptive response seems to be markedly attenuated with aging [9,220,221]. In line with this, a very recent study showed differences in protein abundance between muscles from adults and older subjects at rest, with a marked increase in those related to slow muscle fibers and a significant decrease in glycolytic or mitochondrial proteins in elders. In contrast, the redox state in vastus lateralis muscle was maintained at rest, but a clear disruption was observed following exercise in older subjects. This alteration includes an increase in the oxidation of various cytosolic and mitochondrial proteins and a decrease in protein abundance. Specifically, a significant increase in the number of oxidized cysteine residues was observed in muscles from older subjects, reflecting either an exacerbated increase in ROS production related to exercise or even a diminished antioxidant system. Although the impact of exercise-related redox disruption with aging is not definitively clear, it likely will contribute to a compromised muscle function and probably plays a role in the attenuation of adaptation to exercise [222]. 

PGC-1α is a key regulator of mitochondrial integrity, function, and biogenesis. Several studies have reported that regular endurance exercise, independent of load and intensity, induces gene expression of this factor in skeletal muscle derived from aged animals and from old subjects [223]. Furthermore, lifelong exercise training seems to prevent mitochondrial fragmentation associated with age in skeletal muscle of mice by suppressing mitochondrial fission protein expression in a PGC-1α dependent manner [224]. Moreover, a very recent study carried out in elderly men reported increased levels of different proteins associated with biological aging including PGC1-α after 12-week resistance training. This evidence supports the beneficial effects of exercise on mitochondrial protein and enzymatic function impaired by aging [225]. 

An increase in the transcription factor, Nrf2, considered as the central regulator of intracellular antioxidant response, seems to be another mechanism involved in the response to exercise in an aged subject [7]. A very recent study carried out in aged mice indicated that long-term exercise intervention increased the mRNA expression of Nrf2 in skeletal muscle and improved mitochondrial quality by regulating Drp-1-dependent mitochondrial fission. This resulted in the attenuation of sarcopenia phenotypes in vivo [226]. Growing evidence illustrates the increased expression of Nrf2, which promotes protection against ROS-induced skeletal muscle damage produced by physical activity [227,228]. In addition to its antioxidant function, Nrf2 seems to orchestrate anti-inflammatory processes in skeletal muscle [229]. It is worth mentioning that Nrf2 activation by exercise does not seem to only occur in skeletal muscle, but rather it extends to a systemic level. In line with this, Done and colleagues showed that Nrf2 activity is attenuated in response to exercise in peripheral blood mononuclear cells derived from older adults [230]. 

In addition to the antioxidant role of exercise, another main mechanism by which exercise training reduces age-related functional deterioration is by diminishing muscle inflammation and promoting anabolism, resulting in an increase in muscle protein synthesis [231]. In line with this, it has been previously reported that a multicomponent exercise program in frail obese elderly subjects, but not diet-induced weight loss, down regulated mRNA expression of markers of inflammation such as IL-6 and TNF-α, which are linked to muscle catabolism. Moreover, increased mRNA expression of an anabolic factor, mechano-growth factor of skeletal muscles, was associated with positive effects on functional status [232].

Furthermore, physical activity exerts systemic anti-inflammatory effect. In line with this, a very recent meta-analysis of randomized controlled trials reported that physical exercise resulted in a reduction in the systemic concentration of different inflammatory markers (IL-6, TNF-α and CRP) in middle aged and older subjects [233]. Similar observations were reported in postmenopausal women [234]. Furthermore, a very recent study demonstrated how a moderate-intensity aerobic physical exercise program carried out over 12 weeks reduced the resting expression of inflammasome constituents (NLRP3 and TLR4) and levels of downstream effectors (IL-1β, TNFα, and IL-6) in older women [235].

Additionally, higher levels of physical activity, besides the reduction in pro-inflammatory cytokines, were also associated with higher levels of the anti-inflammatory mediator, adiponectin, and IL-10 [236]. Furthermore, Lavin and colleagues reported that lifelong exercise partially prevented an age-related pro-inflammatory milieu both at the systemic level and at the local level in muscle and maintained the acute inflammatory response observed in young exercising men [237]. 

Finally, it is important to highlight that the impact of physical activity on inflammatory response depends on exercise modality, intensity, and frequency as well as on subject’s characteristics [236]. For example, resistance training attenuated TNF-α expression in aged skeletal muscle, which may result in alleviating muscle changes related to the aging process [214]. Meanwhile, Abd El-Kader and colleagues, showed that the impact of 6 months of aerobic exercise on modulating inflammatory cytokines and immune system response among the elderly was more appropriate than resistance exercise training [238]. Although there is accumulating evidence in older subjects related to the anti-inflammatory effect of exercise due to its influence on cytokine concentrations, some research has failed to reach such positive conclusions [233]. In this sense, Ziegler and colleagues showed that although long-term resistance training enhanced muscle strength and mass, it did not show any effect on resting- or exercise-induced muscle inflammation markers [239]. Moreover, prolonged progressive resistance training had no influence on IL-6, IL-8 and TNF-α circulating levels in either frail or prefrail older adults, despite increased circulating levels of these cytokines being associated with lower strength gains during resistance exercise training [240].

Another benefit of exercise includes stimulating the release of muscle myokines. These muscle factors promote a healthy anti-inflammatory environment, resulting in a decrease in the loss of both muscle function and mass related to the aging process [113]. For example, the PGC-1α-dependent myokine, irisin, which is induced by physical activity, improves mitochondrial function and decreases ROS production [241]. In addition, irisin seems to protect skeletal muscle against metabolic stresses, including oxidative stress, but the mechanism is not well-known [242]. In aged humans, serum irisin levels increased in response to exercise. Furthermore, circulating irisin was significantly decreased in those subjects presenting strength and muscle loss at the end of the study [243] suggesting irisin as a possible marker for improved physical performance in elderly individuals. Furthermore, Mafi et al. showed that the improvement in skeletal muscle strength provided by 8-week resistance training in sarcopenic older adults was related to decreased myostatin and increased follistatin serum levels [244].

In humans and rodents, an age-dependent reduction in the levels of apelin was reported. This exerkine is induced by muscle contraction and is positively associated with the beneficial effects of exercise in older persons. In fact, mice deficient in apelin or its receptor (APLNR) showed dramatic alterations in muscle function with increasing age [98]. 

### 4.2. Cardiovascular System 

The clinical improvement in functional status in the elderly driven by elevated physical activity or exercise performing could be contributed to by the beneficial impact produced by these lifestyle attitudes/interventions on cardiovascular system. Different studies have demonstrated the beneficial effects of exercise on both age-related endothelial dysfunction and arterial stiffening [245]. In this sense, preserved endothelial function in addition to a reduction in age-related arterial stiffness was observed in older subjects who underwent habitual aerobic exercise when compared to sedentary adults [246,247]. 

Available studies identified the diminution of oxidative stress and inflammation as possible mechanisms by which exercise may prevent and/or reverse endothelial dysfunction and arterial stiffness related to the aging process [245]. The above-mentioned benefits of exercise on lowering muscle oxidative stress and inflammation associated with aging might be extended to vascular tissue (Figure 3). In fact, voluntary wheel running (10–14 weeks) by old mice decreased aortic expression of different inflammatory markers (IKK-NF-κB activation, IL-1 and IL-6, IFN-γ, and TNF-α), adventitial and perivascular T cell and macrophage infiltration, and reversed impaired nitric oxide-mediated endothelium-dependent dilation [248].

Furthermore, data obtained from human studies have reported a decreased content of nitrotyrosine, a marker of nitrosative stress, in endothelial cells derived from exercising elderly subjects when compared to the sedentary peer group. In addition, exercise was related to reduced endothelial expression of p47(phox) subunit of the oxidant enzyme, NADPH oxidase, and the redox-sensitive inflammatory transcription factor, NF-κB (p65 subunit). By contrast, the expression of SOD2 and the activity of endothelium-bound extracellular SOD (SOD3) were greater in the exercising group of older subjects [249]. On the other hand, different studies from animal models and humans suggested that exercise training increases NO production, which may play a causal role in the reduction in arterial stiffness risk [250]. Shimomura et al. further supported this observation. A reduction in ADMA elicited by aerobic exercise increased circulating NOx (a surrogate of NO production) and was associated with a decrease in arterial stiffness in both middle age and older men and women [251]. In addition, age-related BH4 deficiency in soleus muscle arterioles was restored by exercise. BH4 restoration prevented eNOS uncoupling and stimulated NO availability [252].

It is noteworthy to mention the crosstalk between muscle derived myokines, induced by exercise, and the vascular bed [96]. In line with this, different studies have suggested that irisin may regulate vascular endothelial function [253]. Furthermore, the study carried out by Fujie and colleagues further supports these observations, since they reported that after 8 weeks of aerobic exercise training, plasma apelin concentrations increased along with plasma NOx levels in middle-aged and older subjects. Meanwhile, a concomitant decrease in arterial stiffness was detected in these subjects [254]. 

However, although age-related micro and macrovascular dysfunction is reversed/prevented by aerobic exercise in men, these positive effects in women are not consistently shown. In this sense, Santos-Parker and colleagues showed in healthy non-obese estrogen-deficient postmenopausal women that despite the fact that aerobic exercise was associated with lower circulating levels of CRP and oxidized low-density lipoprotein compared with the sedentary postmenopausal group, these systemic markers were not correlated with both micro and macrovascular mediated dilations. Those results suggest that aerobic exercise does not protect against age-related forearm micro and macrovascular endothelial dysfunction in this group of women [255].

The effect of exercise is not only limited to alleviating age-related inflammation and oxidative stress, but increases vascular response to external stressors [104]. Aerobic exercise (voluntary wheel running) has been shown to prevent the negative impact of age and Western diet on vascular dysfunction across the lifespan in mice. This protective effect seems to be mediated by the alleviation of vascular mitochondrial oxidative stress and inflammation [256].

On the other hand, despite exercise being accepted as a positive inducer of mitochondrial biogenesis and function in skeletal muscle, its impact on vascular mitochondria remains elusive [257]. Preclinical studies seem to point to vascular mitochondrial function improvements as a possible mechanism underlying the protective effect of exercise on vascular function. This observation is supported by the fact that chronic aerobic exercise enhanced protein expression of the master regulator of mitochondrial biogenesis, PGC-1α, in aorta derived from old animals. The preservation of mitochondrial function by exercise was marked by reduced oxidative stress formation and mitochondrial swelling [258]. Positive changes in mitochondrial health were accompanied by an improvement in endothelium-dependent relaxation [251]. 

Finally, and as mentioned above, Nrf2 plays a key role for redox adaptation to exercise. This latter is also important at the vascular level, since the downregulation of this transcription factor seems to be related to age-associated vascular dysfunction which, in turn, is ameliorated after short term pharmacological activation of Nrf2 with sulforaphane [132]. In this sense, exercise has been shown to increase Nrf2 expression in mouse cardiac fibroblasts [259] and in human peripheral blood mononuclear cells [230].

## 5. Conclusions 

Healthy and successful aging is mainly determined by good functional status in advanced age. In this sense, in an increasingly aged population, trying to identify pathways underlying markers of unhealthy phenotypes seems mandatory. Functional status is compromised by aging-related alterations in the muscle, but also in other organs and systems such as the cardiovascular system. Aging-related impairment of skeletal muscle function and strength precedes the appearance of muscle mass loss, suggesting that the impact of aging affects both muscle quantity and quality. This impact is related to deficient proteostasis with inefficient anabolic pathways of protein synthesis, while protein degradation can be augmented or improperly regulated, leading to misfolded or damaged protein accumulation. Impaired mitochondrial function, increased oxidative stress, reduced antioxidant response, inflammation, and myokine malfunction are also associated with the structural and functional impairment of skeletal muscle with aging (Table 1). Arterial stiffness and endothelial function stand out as the main cardiovascular alterations related to aging. These modifications of vascular health are also associated with increased systemic and vascular oxidative stress and inflammation (Table 2). The combination and expression of these multisystem impairments (skeletal muscle and cardiovascular system), among others, underlie frailty, a geriatric syndrome that is independent but is related to others such as sarcopenia, which warns of an important risk for disability and other negative outcomes in older adults. Some lines of evidence show a pathogenic link between these two systems that could explain its usual concomitant involvement.

Physical activity and exercise training appear as two of the modifiable determinants of functional outcomes in the elderly. High levels of physical activity are related to a lower risk for frailty and other negative functional outcomes in older people. Furthermore, exercise interventions not only prolong functional abilities and retard frailty in the course of aging, but also recover functional performance when prescribed to older populations. Although all exercise intervention modalities have shown beneficial effects, specific prescriptions could improve the functional results in specific older subjects. For this personalized concept of exercise intervention, a deep knowledge of the mechanisms responsible for the therapeutic effects of exercise is required to provide the optimal dose for each improvable deficit. In this sense, physical activity/exercise enhances antioxidant response through Nrf2, decreases pro-inflammatory signals, and promotes activation of anabolic and mitochondrial biogenesis pathways in skeletal muscle. Additionally, exercise reduces inflammatory cytokines and oxidative stress markers at the systemic level, but also improves endothelial function and arterial stiffness through reducing inflammatory and oxidative damage signaling in vascular tissue together with an increase in antioxidant enzymes and NO availability (Figure 4). However, future research is needed to determine the specific mechanisms involved in different types of exercise programs and to explore the specific requirements in the different phenotypes of human aging.

## Figures and Tables

**Figure 1 ijms-23-08713-f001:**
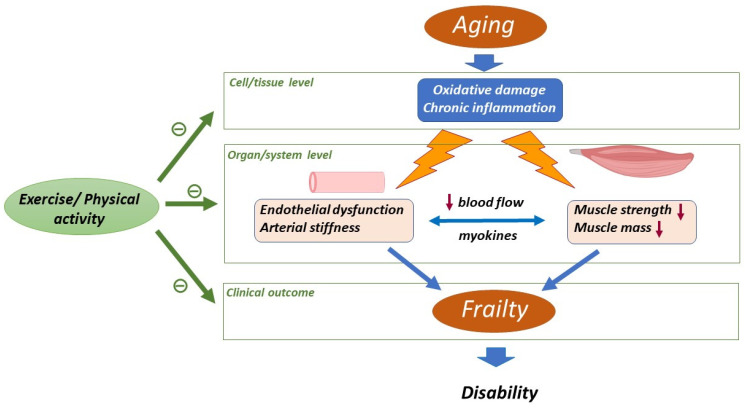
Positive effects of physical activity/exercise on aging-related functional outcomes are evidenced at different levels.

**Figure 2 ijms-23-08713-f002:**
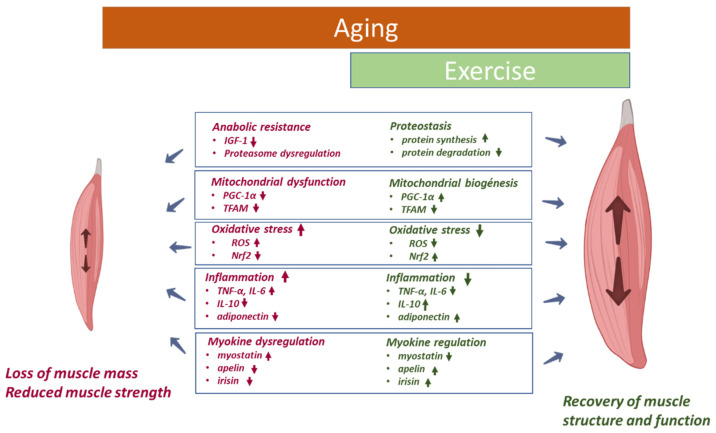
Exercise modulates different signaling pathways affected by aging in skeletal muscle. Aging is associated with muscle mass loss and a reduction in muscle strength resulting from inefficient pathway of anabolic resistance (characterized by decreased insulin-like growth factor (IGF-1) and proteasome dysregulation and mitochondrial dysfunction, specifically reduced expression of factor peroxisome proliferator-activated receptor γ coactivator-1α (PGC-1α) and transcription factor A mitochondrial (TFAM). In addition, increased oxidative stress resulting from the imbalance of increased reactive oxygen species, ROS, and reduced nuclear erythroid-2 like factor-2 (Nrf2), increased chronic low-grade inflammation (increased proinflammatory cytokines and decreased anti-inflammatory cytokines), jointly with myokine dysregulation play a key role in muscle alteration observed with aging. Exercising results in improved proteostasis regulation where protein synthesis increased while protein degradation is reduced. On the other hand, exercise increased mitochondrial biogenesis, reduced age-related oxidative damage, diminished chronic inflammation, and improved myokine profile, which consequently improves muscle structure and function.

**Figure 3 ijms-23-08713-f003:**
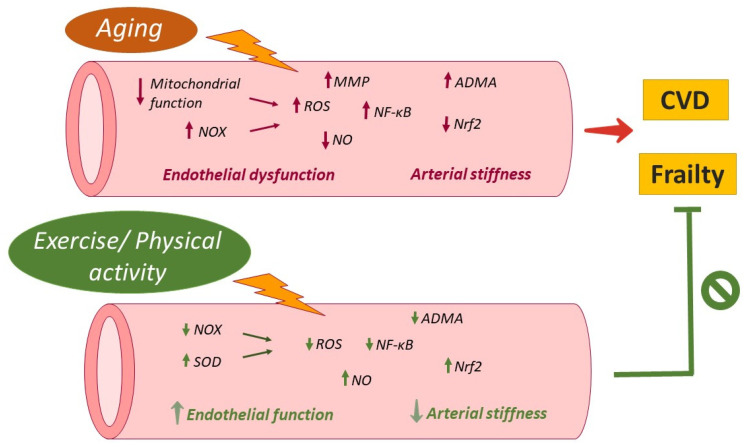
Physical activity/exercise improves different mechanisms through which aging deteriorates vascular function. Endothelial dysfunction and arterial stiffness are two specific vascular phenotypes of vascular aging. The underlying mechanisms of vascular alteration associated with aging includes decreased mitochondrial function and increased expression of vascular NOX, both leading to increased ROS production and further reducing NO availability. Additionally, decreased antioxidant response mediated by Nrf2, increased MMP, reduced SOD activity, enhanced inflammatory mediators, NF- κB, and increased levels of the NO synthase endogenous inhibitor, ADMA, contribute to increasing the risk of developing cardiovascular disease and age-related frailty. Exercise/physical activity improves endothelial function and arterial stiffness through reducing inflammatory and oxidative damage signaling in vascular tissue together with an increase in antioxidant enzymes and NO availability. These improvements prevent or delay the onset of frailty and decrease clinical cardiovascular disease. ADMA: asymmetric dimethylarginine, CVD: cardiovascular disease, NF-κB: nuclear transcription factor-kappa B, NO: nitric oxide, NOX: nicotinamide adenine dinucleotide oxidase, Nrf2: nuclear erythroid-2 like factor-2, ROS: reactive oxygen species, SOD: superoxide dismutase.

**Figure 4 ijms-23-08713-f004:**
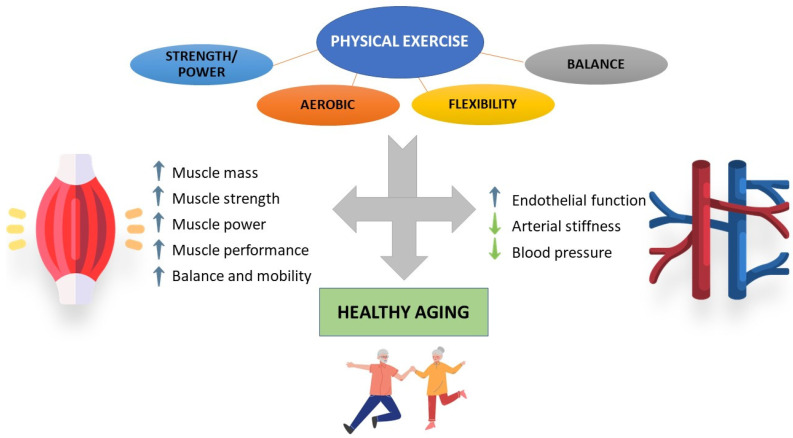
Physical exercise types as promoters of healthy aging. Different modalities of physical exercise (strength/power, aerobic, flexibility, and balance) exert benefits on multiple clinical variables through improving muscular and vascular functions. The expression of these benefits is associated with a healthy aging phenotype.

**Table 1 ijms-23-08713-t001:** Oxidative stress-related pathways influenced by aging and exercise.

Biological System	Signaling Pathway	Tissue Effect	Evidence in Animals	Evidence in Humans	Effect of Exercise
**Skeletal muscle**	Increased ROS/RNS	Cellular damage Defective redox signaling Increased proteolysis	SOD-1 KO mice lost muscle mass and function [77]	Signals of increased oxidative stress was related to functional outcomes in the elderly [26]	Increased catalase expression in trained older subjects [218] Exercise increased SOD and GPX [219] Decreased oxidative stress by exercise was associated with muscle size and strength in older individuals with COPD [217]
	Reduced PGC-1α	Reduced mitochondrial biogenesis Mitochondrial dysfunction	Associated with sarcopenia in mice [59]	PGC-1α was related to reduced physical activity and muscle atrophy in older humans [57,61]	Endurance exercise induced PGC-1α expression in skeletal muscle in aged animals and humans [223] 12-weeks resistance training increased PGC-1α levels in elderly men [225]
	Reduced Nrf2	Increased oxidative stress	Reduced muscle strength and increased fatigue in mice [75,76]	Positive association of Nrf2 with gait speed in older subjects [82]	Long-term exercise increased Nrf2 expression in aged mice related to attenuation of sarcopenia phenotype in vivo [226] Physical activity increased Nrf2 expression promoting protection against ROS-induced damage in skeletal muscle [227,228]
**Vascular system**	Increased ROS/RNS	Reduced NO availability	Inhibition of peroxynitrite normalized vasorelaxation in resistance arteries of aged rats [122]	Increased superoxide anion and peroxynitrite were detected in aged human vessels with defective endothelial vasodilation [121]	Expression of SOD2 and activity of SOD3 were greater in exercising vs. sedentary older subjects [249]
	Uncoupled eNOS/BH_4_ deficiency	Reduced NO availability Increased ROS production	Reduced levels of BH4 were associated with impaired endothelial vasodilation in aged rat arterioles [123]	Vascular reduction in BH4 was related to arterial stiffness and endothelial dysfunction in postmenopausal women [124]	Old age reduced and exercise training restored levels of BH4 in rat soleus feed arterioles related to improved flow-mediated dilation [252]
	Increased NOX	Increased ROS production	Age-dependent increase in blood pressure, cardiomyocyte hypertrophy, coronary remodeling and cardiac fibrosis was associated with increased NOX2 activity [130] Attenuation of NOX activity improved endothelial dysfunction in aged coronary arteries [126]	NOX was overexpressed in arteries from older subjects while NOX inhibition improved endothelial vasodilation [121]	Exercise was related to reduced endothelial NOX expression in elderly subjects [249]
	Reduced Nrf2	Defective antioxidant response Increased oxidative damage	Increased oxidative stress in hearts from old mice correlates with Nrf2 dysregulation and is reversed by sulforaphane [81]	Short-term pharmacological activation decreased age-related impairment of endothelium-dependent and ROS-induced vasodilation in rat and human vascular tissues [132]	Exercise increases Nrf2 expression in mouse cardiac fibroblasts [259] and in human peripheral blood mononuclear cells [230]

BH_4_: tetrahydrobiopterin; COPD: chronic obstructive pulmonary disease; eNOS: endothelial NO synthase; GPX: glutathione peroxidase; KO: knockout; NO, nitric oxide; NOX: NADPH oxidase; Nrf2: nuclear factor erythroid 2-related factor 2; PGC-1α: peroxisome proliferator-activated receptor-γ coactivator-1α; RNS: reactive nitrogen species; ROS: reactive oxygen species; SOD: superoxide dismutase. Reference list numbers of the supportive literature are in brackets.

**Table 2 ijms-23-08713-t002:** Inflammation-related pathways influenced by aging and exercise.

Biological System	Signaling Pathway	Tissue Effect	Evidence in Animals	Evidence in Humans	Effect of Exercise
**Skeletal muscle**	Increased pro-inflammatory cytokines	Muscle inflammation (increased NF-κB) Increased proteolysis	Blockade of TNF-α prevents sarcopenia in aged mice [82]	Increased TNF-α and IL-6 correlates with muscle mass loss in older subjects [90]. Associated with poor physical performance and reduced muscular strength in older subjects [85]	Multicomponent exercise program downregulated expressions of IL-6 and TNF-α in frail obese elderly subjects [232] Physical activity resulted in a reduction in systemic concentrations of IL-6, TNF-α, CRP in middle aged and older subjects and postmenopausal women [233,234] Moderate-intensity aerobic exercise reduced the expression of inflammasome constituents (NLRP3, TLR4) and IL-1β, IL-6, and TNF-α [235]
	Reduced anti-inflammatory cytokines	Muscle inflammation	Muscle overexpression of IL-10 was associated with a low level of muscle inflammation and insulin resistance [94] IL-10 KO mice was proposed as a model of frailty with reduced muscle strength [95]	Increased IL-6/IL-10 ratio in older subjects with sarcopenia [92]	Physical activity was associated with higher levels of IL-10 and adiponectin [236]
	Myokine alteration	Apelin increases mitocondriogenesis and reduces inflammation Myostatin inhibits muscle synthesis and promotes muscle catabolism and is inhibited by follistatin Adiponectin increases mitochondrial function and augments oxidative fibers Irisin improves mitochondrial function decreases ROS production and protects skeletal muscle from metabolic stresses	Apelin restoration prevented muscle wasting in aged mice [92] Adiponectin signaling improves skeletal muscle function in aged mice [97]	Elevated myostatin has been related to sarcopenia in humans [43] Circulating irisin was decreased in older subjects losing muscle strength [243]	Serum irisin levels increased in response to exercise in aged humans [243] Improvement in muscle strength by resistance training was related to decreased myostatin and increased follistatin in sarcopenic older adults [244] Production of apelin in response to muscle contraction contributes to the positive feedback of physical activity and muscle function [98]
**Vascular system**	Activated NF-κB	Vascular inflammation Transcription of inflammatory cytokines and mediators of inflammation Vascular remodeling	Inhibition of vascular NF-κB improved blood flow regulation and decreased systemic inflammation [126]	Enhanced activation of NF-κB in vessels from aged humans which correlated with endothelial dysfunction [121]	Voluntary wheel running by old mice decreased aortic NF-κB activation [248]
	Increased pro-inflammatory cytokines	Endothelial dysfunction Vascular remodeling	Abrogation of inflammasome (NLRP3) preserved cardiac function in old mice and increased the lifespan [137] Anti-inflammatory cytokine, IL-37, improved vascular endothelial function in old mice [142]	CRP and IL-6 have been positively related to aortic stiffness and inversely correlated to endothelial function in older adults [125]	Voluntary wheel running by old mice decreased aortic expression of inflammatory cytokines and macrophage infiltration and reversed impaired NO-mediated endothelial dilation [248]
	Myokine alteration	Irisin improves vascular function Apelin could increase NO production		Higher concentrations of irisin were detected in centenarian people without CVD [113]	Aerobic exercise training increased apelin concentrations along with higher NO production and lower aortic stiffness in middle-aged and older subjects [254]

CRP: C-reactive protein; CVD: cardiovascular disease; IL: interleukin; KO: knockout; NF-κB; nuclear factor-κB; NLRP3: NOD-like receptor protein-3; NO, nitric oxide; ROS: reactive oxygen species; TNF-α: tumor necrosis factor-α; TLR4: Toll-like receptor-4. Reference list numbers of the supportive literature are in brackets.

## Data Availability

Not applicable.

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
