# Peer review of "Effect of Physical Activity/Exercise on Oxidative Stress and Inflammation in Muscle and Vascular Aging"

_ijms, 2022, doi:10.3390/ijms23158713_

Round 1
Reviewer 1 Report
In this article entitled “Effect of physical activity/exercise on oxidative stress and inflammation in muscle and vascular aging” the authors provide an up-to-date review of the possible beneficial effects of physical activity in reducing the cardiovascular disease burden. The authors explore the main mechanisms of the age-related progressive disfunction of muscular and cardiovascular systems, with a specific focus on the molecular and cellular pathways involved in oxidative stress and inflammation. They provide a sum of the evidence about the effects of physical activity and exercise in counteracting the unhealthy aging phenomenon leading to frailty. This is an interesting topic, with many implications for clinicians, and the authors should be complimented for their effort in describing so precisely a great number of molecular pathways and pathological mechanisms.
The article is of potential interest, is well written and the figures are clear. I however suggest some revisions to improve its quality and to make it more “appeal” for the readers. Here are my comments:
1. The abstract should contain a final sentence clearly disclosing the purpose of the review.
2. In a review article what is usually appreciated most by the readers is the immediateness of messages and contents. I therefore suggest adding some tables, at least one summarizing the molecular pathways involved in oxidative stress and inflammation in muscle and vascular aging and another showing the (main) studies investigating these aspects.
3. The article is well written and very detailed. This turns in a quite long paper and long paragraphs. This is mainly due to the abundance of contents and evidence provided, but in some parts of the manuscripts there are very long sentences that could be rephrased and shortened, as well as repetitions that could be avoided. I suggest the authors to see if it is possible to shorten the paragraphs and, in general, the manuscript.
4. The authors argue that endothelial dysfunction and increased stiffness of the large elastic arteries are two main mechanisms of vascular aging. In this context, there are many molecular pathways potentially or certainly involved in microvascular disfunction and a recent review provided a sum of them, also highlighting the possible clinical implication as biomarker or specific target therapy (Rocco, E. et al. Advances and Challenges in Biomarkers Use for Coronary Microvascular Dysfunction: From Bench to Clinical Practice. J. Clin. Med. 2022, 11, 2055. https:// doi.org/10.3390/jcm11072055). Another paper summarized the central role of oxidative stress, inflammation, and fibrosis in determining increased vascular and myocardial stiffness, cardiovascular disease and heart failure, with specific reference to heart failure with preserved ejection fraction (Biasucci, L.M. et al. Novel Biomarkers in Heart Failure: New Insight in Pathophysiology and Clinical Perspective. J. Clin. Med. 2021, 10, 2771. https://doi.org/10.3390/jcm10132771). I suggest the authors to briefly discuss these aspects in paragraph “Cardiovascular system” of chapter “Aging-related changes in skeletal muscle and cardiovascular system” and to cite these two papers.
5. From page 4 line 116 to page 5 line 126 there are several sentences requiring references. Please provide some.
6. There are many mistakes in general in the manuscript about the numeration of the headings and sub-headings. For example, number 1.1 is used for both “Oxidative stress and inflammation, constituents of the background of aging” in section introduction (page 4) and for “Skeletal muscle” in section Aging-related changes in skeletal muscle and cardiovascular system (page 5), but also the other chapters and paragraphs are misnumbered. Please revise and edit the headings and sub-headings of the entire manuscript.
Reviewer 2 Report
Comments to the author:
The Manuscript " Effect of physical activity/exercise on oxidative stress and inflammation in muscle and vascular aging”. The manuscript is not written clearly and orderly, and the cited references need to updated according to appropriate to support the sentences. Also, I have many doubts about the work done in this study;
Major revisions required
Comments:
1- Please correct the heading and subheading nos. it is repeating 1 or 1.1, 1.2 many times. Proper arrangement of manuscript is required
2- PGC 1 alpha and glut also play very important functions in the case of muscles it need to be elaborated
2-Please explain in details about differences in young and ageing and what parameters are responsible for the same
3-How oxygen consumptions are responsible for physical exercise
4-Production of lactic acid
Ex
1- Shen, Singh et al (2022), Antioxidants,11, 1147. Adipocyte-Specific Expression of PGC1α Promotes Adipocyte Browning and Alleviates Obesity-Induced Metabolic Dysfunction in an HO-1-Dependent Fashion.
2-Kanwal, Kanwar, Srivastava, Singh et al, Biomedicines 2022, 10(2), 331. Exploring New Drug Targets for Type 2 Diabetes: Success, Challenges and Opportunities
3- Singh et al (2020), Antioxidants, 9(1) 40. Adipocyte Specific HMOX1 Gene Therapy is Effective in Antioxidant Treatment of Insulin Resistance and Vascular Function in an Obese Mice Model.
Round 2
Reviewer 1 Report
I carefully read the revised version of the paper entitled "Effect of physical activity/exercise on oxidative stress and inflammation in muscle and vascular aging". The authors addressed all my suggestions and were able to provide further references to support their statements. They also checked and edited the headings of paragraphs and provided additional tables. These turned into an improvement in manuscript quality and a better understandability for the readers.
Reviewer 2 Report
It can be accepted in the current form